# Differences in Soil Fungal Communities between Forested Reclamation and Forestry Sites in the Alberta Oil Sands Region

**DOI:** 10.3390/jof9111110

**Published:** 2023-11-16

**Authors:** John. A. Trofymow, Philip-Edouard Shay, Bradley Tomm, Jean A. Bérubé, Tod Ramsfield

**Affiliations:** 1Pacific Forestry Centre, Canadian Forest Service, Natural Resources Canada, 506 West Burnside Road, Victoria, BC V8Z 1M5, Canada; 2Department of Biology, University of Victoria, 3800 Finnerty Road, Victoria, BC V8P 5C2, Canada; 3Canadian Wood Fibre Centre, Canadian Forest Service, 1350 Regent Street, Fredericton, NB E3B 5P7, Canada; philip-edouard.shay@nrcan-rncan.gc.ca; 4Northern Forestry Centre, Canadian Forest Service, Natural Resources Canada, 5320-122nd Street, Edmonton, AB T6H 3S5, Canada; bradley.tomm@nrcan-rncan.gc.ca (B.T.); tod.ramsfield@nrcan-rncan.gc.ca (T.R.); 5Laurentian Forestry Centre, Canadian Forest Service, Natural Resources Canada, 1055 Rue du Peps, Québec, QC G1V 4C7, Canada; jean.berube@nrcan-rncan.gc.ca

**Keywords:** Illumina, pyrosequencing, sporocarp survey, internal transcribed spacer, jack pine, boreal forest

## Abstract

Fungi play key roles in forest soils and provide benefits to trees via mycorrhizal symbioses. After severe disturbance, forest regrowth can be impeded because of changes in fungal communities. In 2013–2014, soil fungi in forest floor and mineral soil were examined by Roche 454 pyrosequencing in undisturbed, harvested, and burned jack pine stands in a forested area near Fort Chipewyan, Alberta. These fungal communities were compared with jack pine, white spruce, and larch stands in Gateway Hill, a nearby certified reclaimed area. In 2014, a more detailed sampling of forestry and reclamation jack pine sites examined fungi in soil fractions using two high-throughput sequencing platforms and a sporocarp survey. The significances of compositional and functional differences in fungal communities between the forested and reclamation sites were assessed using permutation tests of partially constrained ordinations, accounting for confounding factors by variance partitioning. Taxa associated with the forestry area were primarily ectomycorrhizal. Fungal richness and diversity were greater in soils from the reclamation sites and included significantly more pathogenic taxa and taxa with unknown functional properties. Fungal community dissimilarities may have been artefacts of historical legacies or, alternatively, may have resulted from contrasting niche differentiation between forestry and reclamation sites.

## 1. Introduction

Fungal communities are responsible for most of the biochemical cycling in soils of acidic conifer forests [1] and create an intricate link between above- and below-ground communities via functional feedbacks [2]. These communities can be affected by natural and human-caused disturbances [3]. However, it is difficult to generalize about disturbance effects because datasets are frequently incomplete or inadequate [4]. Responses to disturbances are also affected by the differences among individual fungal species and ecological functional groups, e.g., mycorrhizal, saprotrophic, and pathogenic fungi [5,6,7]. Site history can leave lasting legacies affecting microbial community composition [8]. For example, studies of mycorrhizal fungi associated with *Betula* spp. suggest that the succession of fungi may be influenced by the site history (natural stands versus previously treeless sites) [9].

Fire can occur in all forested ecosystems across Canada; however, it is most prevalent in the boreal forests of the west and north [10]. Fire serves many functions, including influencing species composition and age structure, regulating insect and pathogen dynamics, and affecting nutrient cycling and habitat diversity [11]. Forestry and resource management practices increasingly emulate wildfire and other natural disturbance regimes to maintain biodiversity and ecosystem function [12]. In boreal forests, wildfires can affect extensive portions of the landscape, and in the unmanaged areas, the establishment of new trees depends entirely on natural regeneration. In cases of stand-destroying fires that completely consume the forest floor, ectomycorrhizal (EM) fungal species that depend on live tree roots are lost from the site, and their recolonization depends on long-distance dispersal. Following lower-intensity fires, some live trees, forest floor, or understory may remain, and some EM fungal species may be able to persist on the roots of residual tree or woody shrub species or in resistant spores or sclerotia in the forest floor [13]. In the absence of host plants, some pathogenic species will decline in abundance, although many have resistant life stages. Information on the fungal functional groups and associated species in Canadian boreal forests is in the Appendix A.

The effects of silvicultural practices, such as clear-cutting, thinning, and single-species plantations, on sporocarp production have been well documented [14,15,16]. Since EM fungi are associated with tree roots, when trees are removed, EM fungi die over the following one to three years, as the tree root systems die [13]. Some EM fungi species can persist in the soil as spores and can colonize the roots of refuge plants such as woody shrubs, although many species depend on specific host trees and primarily colonize the rooting zone of extant trees [13,17]. Some short-term studies report changes to the species composition alone, with little or no effect on the EM species richness, abundance [18], or inoculum potential [19]. Single-species plantations can increase the incidence and impact of several pathogenic fungi, including species of *Armillaria* [13,20], that have limited spread in high-diversity stands [21]. Partial harvest or disturbance can affect the spread of some pathogenic fungi, which can colonize dead stumps or cut surfaces of live trees and then spread to infect remaining crop trees [22].

Surface mining, one form of deforestation, is more disruptive than forest fire and silvicultural disturbances as all vegetation and surface soils are removed from a site. Mycorrhizal networks are lost, and the taxa capable of growing in the overburden material are typically generic E-strains or disturbance-tolerant *Cenococcum graniforme* [23]. These species and other disturbance-resistant fungi often survive adverse situations in the soil by means of sclerotia or other structures [24]. The removal of woody and leafy material would also cause the demise of lignicolous and other saprobic species and rhizosphere microfungi. Gradually, all the microfungal species that depend on continuous forest, a forest patch, a single tree, or forest shrubs would diminish [25]. When sites are reclaimed and revegetated, the re-establishment of forest mycoflora depends on the type of surface soils used for reclamation and the length of time the soils were stockpiled, the site’s proximity to an established forest, site characteristics (e.g., topography, soil type, and vegetation cover), and the use of other enhancement techniques (e.g., mycorrhizal inoculation of seedlings [18,25]). Mycorrhizal communities in soils from oil sands or other reclamation areas can be depauperate [26], thus affecting tree regeneration.

To facilitate forestry and mining, land tenure systems managed by provincial governments regulate land use for a finite period to companies who are then responsible for resource development and remediation. In the oil sands region of Alberta north of Fort McMurray, bitumen surface mine leases cover a large area, many with active mines (Figure 1). Each lease involves lands in different stages of development and under different land cover: wetlands; recently burned, harvested, or regenerating juvenile and mature forests; recently deforested lands; active mines; and recently reclaimed and revegetated lands. All of these stages and land covers represent opportunities to examine the effects of disturbance type and time since disturbance on soil fungal communities.

Understanding the effects of various disturbance types on fungal communities can help explain differences in productivity of reforested areas following disturbance and has important implications for reclamation and forest management. This study compared the soil fungal communities in different stands in a forestry area accessible along the Fort Chipewyan road with those in forested stands at an established reclamation area (Gateway Hill), about 50 km to the south. Both areas were in the Boreal Mixedwood zone with annual mean temperature 1.5 °C, total precipitation 389 mm; summer temperature 13.7 °C (min 7.2 °C, max 20.2 °C), precipitation 238 mm; and winter temperature −11.9 °C (min −17.2 °C, max −6.5 °C), precipitation 63 mm. [28]. We hypothesized that the soil fungal communities at the Gateway Hill area, which was formed from heavily disturbed soils, would generally be depauperate when compared with those in disturbed and undisturbed stands in the forestry area along the Fort Chipewyan road, which had less disturbed soils and covered a larger spatial extent. We also hypothesized that the largest differences would be for the EM fungal species, most of which are unable to survive without live trees and may depend on root contact for dispersal into regrowing forests. In a preliminary assessment, we examined soil fungal communities over the full range of stand conditions in both areas. This was followed by a more detailed examination of soil fungal communities on transects in two jack pine stands at each area using multiple methods.

## 2. Materials and Methods

### 2.1. Site Descriptions

To compare different disturbances on a similar landscape, sampling took place on an accessible reclaimed area no longer under lease, since access to mine lease lands is strictly controlled, and nearby accessible forestry stands.

The 104 ha Gateway Hill reclamation area is on land that was formerly part of one of the oldest bitumen mines in the region (the Syncrude Mildred Lake Mine). The area was a former overburden dump on which reclamation began in 1982. Initially, reclamation involved placement of 15–30 cm of muskeg peat/sandy loam surface soil mixture directly on the overburden of lean oil sands. After 1983, the approach was changed to placement of subsoils of 60–100 cm of sandy to clay loam and then the surface soil mixture. After 1989, the subsoil was changed to 85–150 cm of mineral/muskeg peat mix [29]. The area was also revegetated, with strips of jack pine (M2, M7 *Pinus banksiana* Lamb.), white spruce (M6, M14 *Picea glauca* (Moench) Voss), or Siberian larch (M8 *Larix sibirica* Ledeb.) progressively planted following reclamation (Figure 1).

Forest stands of varying disturbance history occurred along the road to Fort Chipewyan. These managed forest lands were dominated by jack pine on sandy, nutrient-poor soils. Stands examined included an undisturbed approximately 70-year-old stand (TR4U), a 17-years-post-harvest planted stand (TR2H), and stands affected by the 2011 Richardson fire (Figure 1). Partially and lightly burned stands (TR1P and TR3L) maintained live standing trees, while severely burned stands (TR1B1 and TR1B2) were blanketed with naturally regenerating seedlings and devoid of live mature trees. 

### 2.2. Sample Collection and Processing

In September 2013, single soil collections of surface organic and mineral layers were performed at four Fort Chipewyan road stands (TR1B1, TR2H, TR3L, and TR4U). In September 2014, single soil collections were obtained from tree plantations at the Gateway Hill area, one in each of two white spruce stands (M6 and M14) and one in a Siberian larch stand (M8). As well, sporocarp surveys were conducted, and soil cores were collected at three replicate locations (15 m apart) along 50 m transects in two reclaimed jack pine stands at Gateway Hill (M2 and M7) and two jack pine managed forest stands along Fort Chipewyan road that had been affected by fire in 2011 (TR1P and TR1B2).

In both years, the forest floor depth was recorded, and a trowel, wiped clean between samples, was used to remove the forest floor in a 15 cm × 15 cm square to the surface of the mineral soil, designated the “forest floor” fraction. Mineral soil was then removed from a 15 cm × 15 cm square, approximately 10 cm deep.

In the laboratory, branches, cones, and gravel were removed from the forest floor before soil was freeze-dried and ground to a fine powder using a mortar and pestle. In both years, mineral soil was passed through a 2 mm sieve, with the retained material (mostly roots) designated the “root” fraction and the material passing through the sieve designated the “soil” fraction. In 2014 the material less than 2 mm was further sieved through a 500 µm sieve, and the retained material was designated the “coarse soil” fraction, whereas the material passing through the sieve was designated the “fine soil” fraction. Visible roots were removed from all soil fractions and combined with the “root” fraction. Between samples, the sieves were thoroughly washed, rinsed with 95% ethanol, and allowed to air dry. Each fraction was homogenized, and a subsample (≤50 mL) was frozen, freeze-dried, and ground using a mortar and pestle. All material was stored at −20 °C before DNA extraction.

The 2013 sample “root” fraction was discarded, and the <2 mm soil was not further sieved through a 500 µm sieve. As a result, to allow for comparison across all 11 stands and both years, the preliminary assessment included only data for the “forest floor” fraction, and the 2014 data for the “coarse soil” and “fine soil” fractions were combined to obtain values comparable to the 2013 sample “soil” fraction. To equalize the sampling intensity, the preliminary assessment combined data from single cores collected in 2013 and 2014 with one randomly chosen soil core from each of the four jack pine transects (TR1B2, TR1P, M2, and M7) sampled in 2014.

The more detailed examination of the four 2014 jack pine transects used samples from all four soil fractions and the three replicates at each site (Table 1). In addition to DNA extraction, the transect samples were also analyzed for nutrient concentrations in forest floors and fine mineral soils by the analytical laboratory of the Northern Forestry Centre, Canadian Forest Service (Edmonton, AB, Canada; Appendix A). Available nitrate (NO_3_^−^) and ammonium (NH_4_^+^) were measured using 2 mol/L potassium chloride extraction followed by phenate blue colorimetric detection, and extractable phosphorus (P) was measured using the Bray method [30]. Dry combustion using a TruSpec^®^ CHN Analyzer followed by thermal conductivity detection (Leco Corporation, Saint Joseph, MI, USA) was used to assess total nitrogen (N). The chemistry of the 2013 soil samples was not measured.

### 2.3. Sporocarp Survey on Jack Pine Transects (2014)

Sporocarp surveys were conducted in 2014 along 50 m transects in the four jack pine stands. All fungal sporocarps within 1 m on each side of the central line (100 m^2^ total area) were counted and categorized by morphology, and voucher specimens were photographed and collected. Samples of tissue (five to six pieces of approximately 4 mm^3^ each) from the vouchers were submitted to the Biodiversity Institute of Ontario (Guelph, ON, Canada) for DNA extraction, polymerase chain reaction (PCR) on the internal transcribed spacer (ITS) region of the ribosomal DNA primers, and DNA sequencing [31]. Specimen metadata, images, and DNA sequences were accessioned in the barcode of life database [32], under the SVMFA project, and voucher specimens were accessioned in the Department of Agriculture, Victoria, Forest Pathology fungal herbarium. The DNA sequence data were subjected to basic local alignment search tool (BLAST) searching [33] in both the GenBank and the UNITE [34] databases, and identifications were based on judgment of the best sequence matches and morphological features.

### 2.4. Soil DNA Processing and Sequencing

DNA was extracted from 0.25 g subsamples using PowerSoil^®^ DNA Isolation kits (MoBio Laboratories Inc., Carlsbad, CA, USA). Soil fungal communities were targeted using polymerase chain reaction (PCR) amplification of ITS regions from the extracted DNA (5 µL per sample). Amplicons were then sequenced using 454 pyrosequencing (Roche/454 Life Sciences, Basel, Switzerland). Fungal community assemblages were obtained for all samples using pyrosequencing, except for two forest floor and two fine soil samples from the severely burned transect at Fort Chipewyan road (TR1B2), which failed to amplify ITS amplicons. Also, DNA from a jack pine coarse soil sample at Gateway Hill was contaminated with a white spruce coarse soil sample. These samples were omitted from analyses; however, the effects of reclamation status on fungal communities in jack pine soils were not affected by inclusion of this sample in initial analyses. DNA was later re-extracted from stored jack pine coarse soil fractions (six samples for each reclamation and forestry area) and ITS amplicons sequenced using the Illumina MiSeq platform (Illumina, San Diego, CA, USA).

For the 454 DNA pyrosequencing method, ITS regions of the ribosomal DNA fragment (ITS1–5.8S–ITS2) were amplified using the primer constructs, which combined (1) the 454-sequencing adaptor A and a specific 10 bp DNA tag/multiplex identifier (for post-sequencing sample identification) with the fungal specific primer ITS1F [31] and (2) the DNA capture bead anneal adaptor B with the universal primer ITS4 [35] for the emulsion PCR (the adaptors and multiplex identifiers are published by Roche/454 Life Science; for PCR conditions, (see Appendix A for detailed methods).

For the Illumina sequencing method, ITS regions of the ribosomal DNA fragment (ITS1–5.8S) were first amplified in a two-step PCR using the Illumina fusion primers [36]. These primers contained an index sequence for tagging every sequence to a sample. Fifteen forward-indexed sequences were used in combination with 15 reverse-indexed sequences to produce 225 indexed combinations.

All amplicons were then purified using an Agencourt^®^ AMPure^®^ XP magnetic PCR clean-up system (Beckman Coulter, Brea, CA, USA), which eliminated primer dimers and small fragments (smaller than 200 bp using a purification ratio of 0.6:1 for 454 pyrosequenced amplicons and smaller than 325 bp using a purification ratio of 0.7:1 for Illumina sequenced amplicons). The clean PCR amplicons were quantified with the Quant-iT™ Picogreen^®^ dsDNA assay kit (Invitrogen, Eugene, OR, USA). The DNA concentrations were measured with a Fluoroskan Ascent Labsystem fluorometer (Thermo Electron Corporation, Vantaa, Finland), with an excitation wavelength of 486 nm and emission wavelength of 585 nm.

The tagged amplicons samples were then pooled in equimolar amounts of 4 ng DNA per sample. The final quantification of the pool, the verification of the primer artifact removal, and the amplicon quality check were performed with the Agilent 2100 BioAnalyzer system (Agilent Technologies, Santa Clara, CA, USA). The pooled DNA samples (75 ng each) were sent for 454 pyrosequencing to the McGill University and Genome Québec Innovation Centre (Montréal, QC, Canada), which performed the emulsion PCR with the Lib-L GS FLX Titanium PCR kit according to the manufacturer’s instructions (454 Life Sciences, Branford, CT, USA) and unidirectional sequencing on a one-quarter PicoTiterPlate with a GS-FLX Titanium sequencer (454 Life Sciences, Branford, CT, USA). Samples for Illumina sequencing were sent to the Genomics Centre, Centre Hospitalier de l’Université de Québec, Université Laval Research Centre (Québec, QC, Canada), which performed paired-end 300 bp sequencing using MiSeq Reagent Kit v3 (600 cycles) with an Illumina MiSeq system.

### 2.5. Soil DNA Bioinformatic Analyses

A stringent treatment of 454 DNA pyrosequences was executed to prevent the formation of a disproportionate number of fictitious operational taxonomic units (OTUs), our proxies for fungal species [37,38,39]. Analyses were performed by using mothur, v.1.28.0 [40]. Sequences were denoised (using Pyronoise implementation in mothur), then filtered and trimmed (reads shorter than 120 bp, after removal of barcodes, tags, and primers, were discarded; unambiguous positions and a maximum homopolymer length of 9 bp were tolerated).

Dereplication was performed on the full length of each sequence set before construction of clusters with USEARCH v.6.0.307 [41]. Reads were agglomerated into clusters, which formed the OTUs, using a sequence similarity threshold of 97% and the most abundant sequence types as cluster seeds. As no single similarity threshold accurately reflects the species level throughout the fungal kingdom, a cutoff of 3% dissimilarity was selected as a compromise to avoid overestimating the fungal diversity and masking cryptic OTUs [37,42,43]. Representative sequences, which were the most frequent sequence in each OTU, were extracted and then screened against relevant databases using local BLAST v.2.2.28+ [33,41,44]. Twenty-five top best BLAST hits were sought in databases by a BLASTn, setting the minimum identity and query coverage parameters to 80%.

Raw Illumina forward and reverse DNA reads were reassembled using PANDASeq, v2.11 [45]. Resulting sequences were then processed by Illumicut [46], which removed amplification primers and then sequences that were too short or ambiguous (length < 200 bp or at least one ambiguity, respectively). Trimmed and filtered sequences were then dereplicated using mothur unique.seqs command (default parameters [40]). Dereplicated sequences were then screened for long homopolymer chains (>9 bp) using the HomopRemover program [47]. Sequences with long homopolymer chains but occurring more than once were conserved. Sequences with 97% similarity or more were then clustered using VSEARCH software, v2.7.0 [48]. The resulting OTUs were then initially identified using a VSEARCH BLAST search of the GenBank fungi-only database.

For 454 pyrosequenced data, all 2013 and 2014 samples were run through a single bioinformatics analysis run, and sequences with lengths ≤ 100 bp or with ≤10 reads across all samples were removed from the statistical analyses. For Illumina data, sequences with ≤10 reads for the 12 samples from the 2014 jack pine transects were removed before statistical analyses.

### 2.6. Taxonomic Identification of Soil DNA OTUs

The similarity and coverage scores obtained by the BLASTn analysis for each representative ITS sequence allowed us to propose a taxonomic identification for each OTU (Schoch et al., 2012). An overview of fungal diversity covered by the sampling was evaluated using BROCC v.1.1.0 to attribute the best consensus taxon among the top-hits identification returned by the BLASTn screening of the GenBank ITS fungal database (the parameters for taxonomic assignment were set for the ITS genes [49]).

Fungal OTUs associated with known taxa in GenBank databases (minimum of 77% and 80% identity match for 454 pyrosequencing and Illumina sequencing, respectively) were identified, and the reads were pooled when OTUs mapped to the same GenBank accession number. The identified taxa were assigned functional properties according to the trophic status, lifestyles, decay types, and growth forms designated by [7].

### 2.7. Statistical Analyses

All analyses were performed using the R program for statistical computing [50]. The abundances of fungal taxa and functional groups were always standardized according to counts per million reads within each sample. Both the presence/absence of fungal taxa and the number of taxa belonging to functional groups were also assessed to account for the semi-quantitative nature of sequence data [51]. Soil fractions were treated separately as well as jointly.

The preliminary assessments of the effect of reclamation status considered forest floors and mineral soils collected in 2013 and 2014 from the Fort Chipewyan road area (*n* = 6 for each soil fraction) and in 2014 from the Gateway Hill area (*n* = 5 for each soil fraction). To adequately compare mineral soils collected in 2013 with serially sieved samples collected in 2014, the fungal abundances in coarse and fine soils were summed. Analyses using solely coarse or fine soils were comparable to those using summed abundances. The more detailed examination of the four 2014 jack pine transects eliminated the effect of tree species on the impact of reclamation status on soil fungi and increased the sampling intensity, as four separate soil fractions were assessed in each of the three replicate cores from each site (43 samples of a potential 48 samples, with five samples missing due to PCR or contamination issues).

Dissimilarities among soil fungal communities (taxa and functional groups) were first assessed using non-metric multidimensional scaling (NMDS) ordination-based ordering and network analysis (taxa only) of Bray–Curtis distances using the phyloseq package [51]. The effects of reclamation status on soil fungal communities were then assessed using analysis of variance (ANOVA)-like permutation tests (999 permutation) of (partially) constrained ordination analyses using the vegan package [52]. Only correlations greater than 0.5 or less than −0.5 with constrained ordinations axes were described, unless otherwise specified. The fungal taxa were analyzed using constrained correspondence analyses, while the functional groups were analyzed using redundancy analyses (RDAs), based on the binomial nature of the response variables or range distributions when using detrended correspondence analyses [53]. 

When applicable, models in preliminary assessment (2013 and 2014 samples) accounted for tree species, stand stage (seedlings, juveniles, and young), tree status (live versus dead with seedlings), regeneration year, and spatial distance. Models limited to the 2014 jack pine transect samples accounted for the possible confounding effects of regeneration year, forest floor depth, nutrient concentrations in mineral soil, and spatial distance. The principal coordinates of neighborhood matrices (PCNM) of geospatial coordinates of sampling transects were used to assess the effects of spatial distance [54]. Significant PCNM components and nutrients were selected independently by backward and forward stepwise elimination of least-significant factors based on Akaike information criterion-like statistics using the ordistep function [55], followed by the stepwise elimination of non-significant factors by permutation tests in which terms were added sequentially as well as being tested for marginal effects. The significance of each confounding factor was first assessed independently in constrained models before inclusion as conditions in partially constrained ordinations testing the significance of reclamation status. The variance partitioning among the reclamation status, spatial layout, and other factors was calculated according to the methods first described by [56].

Diversity indexes (Chao1, abundance-based coverage estimator (ACE), Shannon, inverse Simpson, and Fisher) were calculated using the estimate_richness function of the phyloseq package [57]. The effects of reclamation status on the richness and diversity indexes were assessed using linear mixed-effects models with the soil fraction as a random effect, followed by post hoc Welch two-sample *t*-tests treating each soil fraction separately.

The effect of the burn intensity on pyrosequenced fungal communities from partially and severely burned jack pine transects at Fort Chipewyan road stand TR1 was also assessed using similar statistical approaches.

## 3. Results

### 3.1. Preliminary Assessment of Forestry and Reclamation Stands (2013–2014)—Pyrosequenced Fungal Communities

Across forestry (Fort Chipewyan road) and reclamation (Gateway Hill) sites (22 samples), 113,811 reads representing 310 fungal taxa were detected, of which 288 were detected in mineral soils and 238 were in forest floors. More taxa were detected in the 10 reclamation area samples (260 total, of which 240 were in mineral soils and 196 were in forest floors) compared with the 12 forestry area samples (184 total, of which 166 were in mineral soils and 139 were in forest floors). *Cenococcum geophilum* Fr. and *Mortierella alpina* Peyronel were ubiquitously detected in all samples. The richness and diversity of soil fungal communities were significantly greater in reclamation sites compared with forestry sites when considering mineral soil or both soil layers jointly, regardless of the diversity index used (*p* < 0.05, Appendix A). The richness, Chao1, ACE, and Fisher indexes of diversity in forest floor samples were significantly greater in reclamation sites compared with forestry sites (*p* ≤ 0.006); however, the Shannon and Simpson indexes of diversity were not (*p* = 0.527 and *p* = 0.862, respectively).

Among detected taxa, 147 were classified as saprotrophs, 84 were classified as symbiotrophs, and 20 were classified as biotrophs, according to the classification of [7]. Symbiotrophs consisted mainly of EM taxa (74), of which fewer were detected in reclamation sites (53) than in forestry sites (56). Only three taxa were classified as arbuscular mycorrhizae and one as an ericoid fungus. Most biotrophs were classified as plant pathogens (17), of which twice as many were detected in reclamation sites (16) than in forestry sites (8). More saprotrophic taxa (37%) were detected in reclamation sites (129) than in forestry sites (82).

Ordering samples based on NMDS of Bray–Curtis dissimilarity analyses of fungal taxa consistently separated reclamation and forestry sites, whether highly variable taxa (Figure 2), the most abundant taxa, or all detected taxa were considered. Assessments of functional groups in mineral soils also separated reclamation from forestry sites; however, two forest floor samples from Gateway Hill clustered with Fort Chipewyan road samples (Figure 3).

The spatial layout accounted for up to 64% of the variance in fungal communities (Appendix A). Spatial PCNM 1 delineated reclamation from forestry sites (r = −0.998, *p* ≤ 0.001) and confounded effects of reclamation status. Therefore, conditioned models testing the effects of the reclamation status did not account for PCNM 1. Other potentially confounding factors accounted for up to 25% of the variance in fungal communities, yet, for the most part, only when considering both soil layers jointly (Appendix A). When treating soil layers independently, associations were detected only between tree species and community composition in forest floors (24.62%; *p* = 0.019; Appendix A). Tree species consistently accounted for variance in fungal communities when considering both soil layers jointly. Tree species accounted for 23% to 46% of the variance associated with reclamation status, in part since tree species varied only at the reclamation area. The stand stage and tree status were associated with community composition, although no association with the regeneration year was detected. The stand stage and tree status effects distinguished severely burned stands with seedlings from stands at other locations.

The reclamation status significantly accounted for the variance in the composition of soil fungal communities, except when considering the presence/absence of taxa in forest floors due to the confounding effects of tree species (Appendix A and Figure 4A–C). More fungal taxa were consistently detected in reclamation sites (63) than in forestry sites (38; Appendix A; Figure 4A–C). A greater proportion of site-specific taxa were EM in the forestry area than in the reclamation area (34% and 13%, respectively). Conversely, the proportion of saprotrophic taxa was greater in reclamation sites than in forestry sites (56% and 39%, respectively). Site-specific plant pathogens were detected only in the reclamation area, namely, *Phoma herbarum* Westend., *Microdochium nivale* (Fr.) Samuels & I.C. Hallett, and *Verticillium leptobactrum* W. Gams. Although the range in the number of taxa among the Fort Chipewyan road forestry sites was large (Figure 4D,F), the limited sampling meant it was not possible to statistically test for site-specific factors. Nonetheless, the mean number of total and saprotrophic fungal taxa in the sites that had been burned in 2011 (137 ± 9.6 SE, 67 ± 4.5 SE) was greater than in the unburned sites (102 ± 6.0 SE, 44 ± 4.5 SE), but the number of EM taxa was the same (39 at burned sites and 40 at unburned sites).

Reclamation status was associated only with the relative abundance of functional groups in mineral soils (Appendix A). The relative abundances of plant pathogens, saprotrophic white rots, and EM white rots were greater at the reclamation area (*p* = 0.005; Figure 4D). The number of plant pathogens, saprotrophs, and taxa with unknown functional properties was greater at the reclamation area, regardless of the soil layer (*p* ≤ 0.005; Figure 4E,F). The animal parasite *Beauveria bassiana* (Bals.-Criv.) Vuill. was detected only at the reclamation area.

### 3.2. Reclamation and Forestry Jack Pine Transects (2014)

#### 3.2.1. Abiotic Attributes

The geospatial coordinates of the sampled transects were broken down into two PCNM components. The regeneration year correlated with PCNM 2 (r = 0.615, *p* < 0.05). The greatest spatial distance between samples was described by PCNM 1, which separated the Gateway Hill area (reclamation sites M2, M7) from the Fort Chipewyan road area (forestry sites TR1B2, TR1P). This spatial component was highly correlated with forest floor depth (r = −0.729, *p* = 0.007) and mineral soil P concentrations (r = 0.986, *p* < 0.001), which were negatively correlated with each other (r = −0.787, *p* = 0.002). The forest floor depths were on average 6.4 times greater in reclamation sites (2.6 cm) compared with forestry sites (0.4 cm; Welch two-sample *t*-test *p* = 0.005; Appendix A). The average P concentration in mineral soils at forestry sites (35.50 μg g^−1^) was 65 times greater than at reclamation sites (0.55 μg g^−1^; Welch two-sample *t*-test *p* < 0.001; Appendix A). Hence, PCNM 1, mineral soil P, and forest floor depth shared substantial portions (12% to 100%) of the variance in fungal communities associated with reclamation status and often confounded analyses of reclamation effects, especially when accounting for their joint effects and/or when treating soil fractions separately. Therefore, the identification of fungal taxa and functional groups affected by reclamation status focused on partially constrained models that omitted these three variables (Table 2).

The regeneration year and other nutrient concentrations were not significantly different between reclamation and forestry sites (Welch two-sample *t*-tests *p* > 0.05; Appendix A), despite greater mean total N in mineral soil at reclamation sites (0.06 vs. 0.45 μg g^−1^). The concentrations of NO_3_^−^ and NH_4_^+^ in mineral soils were highly correlated (r = −0.619; *p* = 0.032). These confounding factors, along with PCNM 2, accounted for up to 35% of variance in functional groups and 28% of the relative abundance of fungal taxa (Table 2).

#### 3.2.2. Pyrosequenced Fungal Communities

Across all soil fractions in jack pine transects at the Gateway Hill (M2 and M7) and Fort Chipewyan road areas (TR1B2 and TR1P) (43 samples), 311 fungal taxa were detected among 161,138 sequences. Of these, 270 fungal taxa were detected in coarse soils, 265 were detected in fine soils, 229 were detected in the forest floor, and 257 were detected in root fractions. Fewer taxa were detected only in roots (two) than in other soil fractions (eight each) or across fine and coarse soils (27). *Mortierella alpina* was the only taxon ubiquitously detected in all samples. The mean richness was significantly greater in coarse soils (69.7 and 105.8 for forestry and reclamation areas, respectively) than in roots (55.7 and 81.7 for forestry and reclamation areas, respectively), regardless of the area (*p* = 0.001). The fungal richness was significantly greater in reclamation sites (258 total, 93.4 ± 4.8 SE mean per sample) than in forestry sites (197 total, 67.5 ± 3.8 SE mean per sample, *p* < 0.05; Appendix A). All diversity indexes, except Shannon’s index (and Simpson’s inverse when treating soil layers separately), were also significantly greater at the reclamation sites (*p* < 0.05; Appendix A). The proportion of locally detected taxa unique to a site was greater in reclamation sites (0.44) than in forestry sites (0.27).

Among the 311 fungal taxa detected, 152 were classified as saprotrophs, 79 were classified as symbiotrophs, and 21 were classified as biotrophs. Symbiotrophs consisted mainly of EM taxa (72). Only two taxa were classified as arbuscular mycorrhizae, and only one was classified as an ericoid fungus. Most biotrophs (19) were classified as plant pathogens. Among the saprotrophs, six taxa were deemed yeasts, six were facultative yeasts, three were brown rot fungi, and three were white rot fungi. Overall, more taxa deemed saprotrophic and plant pathogens were detected in reclamation (129 and 16, respectively) versus forestry sites (92 and 9, respectively). Slightly more EM taxa were detected in reclamation (58) than in forestry sites (54).

Network analyses of the relative abundance of fungal taxa consistently clustered reclamation samples separately from forestry samples (Figure 5). Clustering based on NMDS also separated reclamation from forestry sites, whether treating soil fractions jointly (Figure 2) or separately. When only highly variable taxa were assessed, the most abundant taxa or all detected taxa did not affect the clustering outcomes. The analyses of trophic functions also consistently clustered reclamation samples separately from forestry samples, except for one reclamation forest floor sample (transect M2, replicate 2; Figure 3). *Wilcoxina rehmii* Chin S. Yang & Korf composed a substantial portion of the fungi in this forest floor sample (325 of 1049 total reads) and contributed to a high proportional abundance of EM fungi.

The reclamation status significantly affected the composition of the fungal communities in each soil fraction, regardless of the confounding factors except PCNM 1, mineral soil P, and forest floor depth (Table 2 and Appendix A). More fungal taxa were consistently detected in both roots and coarse soils in reclamation sites (62) than in forestry sites (49; Appendix A; Figure 6A,B). A greater proportion these taxa were deemed EM and saprotrophs in forestry sites (22% and 47%, respectively) than in reclamation sites (16% and 34%, respectively). Conversely, a greater proportion of the identified taxa were plant pathogens in reclamation sites (11%) than in forestry sites (2%). Furthermore, when roots and coarse soils were independently assessed, *Colpoma* sp. PDD 91607 and *Devriesia pseudoamericana* J. Frank, B. Oertel, Schroers & Crous were also associated with forestry sites, while *Nectria berolinensis* (Sacc.) Cooke and *Phoma* spp. were associated with reclamation sites (*p* < 0.001).

The total number of taxa belonging to functional groups was also affected by the reclamation status (Table 2); however, the number of taxa was not affected in forest floor samples when differences in NO_3_^+^ were accounted for (Appendix A). The numbers of saprotrophs, plant pathogens, arbuscular mycorrhizae, and fungi with unknown fungal properties were significantly greater in reclamation sites (*p* < 0.001; Figure 6D). Mycoparasitic fungi (i.e., *Hypomyces australis* (Mont.) Höhn.) were detected only in forestry sites, mainly in coarse and fine mineral soils. Conversely, animal parasites (i.e., *Beauveria bassiana*) were detected only in reclamation sites. The relative abundance of functional groups was primarily affected by the reclamation status in coarse and fine mineral soil fractions (Table 2 and Appendix A). The relative abundance of saprotrophs in coarse soils was greater, and that of saprotrophic yeasts was lesser, in forestry sites than in reclamation sites, regardless of the model conditioning (*p* < 0.001; Figure 6C). The relative abundance and number of taxa deemed lichenized fungi were also greater in soils in forestry sites than in reclamation sites (*p* < 0.001). The effects of the reclamation status without confounding factors are visualized in Appendix A.

#### 3.2.3. Burn Intensity Effects from Pyrosequenced Fungal Communities

Fire intensity (forestry site TR1P versus TR1B2) did not affect the fungal richness and generally did not affect the diversity (*p* > 0.05) but was significantly associated with the presence/absence of fungal taxa (*p* = 0.014). The same number of taxa were specifically associated with either partially or severely burned stands (43), but a greater proportion of these taxa were EM in the partially burned site (42%) than in the severely burned site (16%), while the reverse was true for saprotrophs (32% in the partially burned site versus 49% in the severely burned site). The relative abundance of all detected EM taxa was also greater in the roots and coarse soil fractions of the partially burned site than in the severely burned site (*p* < 0.001).

#### 3.2.4. Illumina-Sequenced Fungal Communities

Illumina sequencing of DNA from the 12 coarse soil samples resulted in 596,111 sequences clustered to 895 fungal taxa. More taxa were detected in reclamation sites (784 total, 387 ± 60 SE per sample mean) than in forestry sites (379 total, 202 ± 15 SE per sample mean; Welch two-sample *t*-test for means *p* = 0.026). All indexes of fungal diversity were also greater in reclamation sites than in forestry sites (*p* < 0.05). The proportion of locally detected taxa unique to a site was greater in reclamation sites (0.66) than in forestry sites (0.29). *Mortierella alpina* strains composed 3 of the 14 taxa ubiquitously detected in all samples (Appendix A).

Among the 895 fungal taxa detected, 352 were classified as saprotrophs, 126 were classified as symbiotrophs, and 97 were classified as biotrophs. Symbiotrophs consisted mainly of EM taxa (108). Only two taxa were classified as arbuscular mycorrhizae, and nine were classified as ericoid fungi. Most biotrophs were classified as plant pathogens (66), animal parasites (13), and mycoparasites (13). Among saprotrophs, seven taxa were deemed yeasts, 36 were facultative yeasts, seven were brown rot fungi, and five were white rot fungi. Overall, more taxa deemed saprotrophic and plant pathogens were detected in reclamation sites (323 and 59, respectively) than in forestry sites (169 and 25, respectively). Slightly fewer EM taxa were detected in reclamation sites (75) than in forestry sites (77).

Network analyses of Bray–Curtis distances clustered samples from reclamation sites separately from forestry sites (Figure 5). Samples from reclamation sites were linked only to samples collected along the same transect, while samples from different transects in the forestry sites were interlinked (Figure 5). Clustering samples based on NMDS also grouped reclamation samples separately from forestry samples, whether only taxa with highly variable abundance (Figure 2), the most abundant 250 taxa, or all detected fungi were considered. Fungal communities were dominated by taxa with saprotrophic and EM properties across sites, yet when an unconstrained assessment of the functional properties was conducted, it distinctly grouped each transect and separated reclamation from forestry sites (Figure 3).

The reclamation status accounted for more variance in the fungal community composition (29%) than in the relative abundance of these taxa (16.44%; Table 2). More fungal taxa were consistently detected in reclamation sites (115) than in forestry sites (81; Appendix A; Figure 7A,B). A greater proportion these taxa were deemed EM and saprotrophs in forestry sites (33% and 38%, respectively) than in reclamation sites (12% and 30%, respectively). Conversely, a greater proportion of the identified taxa were plant pathogens and fungi with no known functional properties in reclamation sites (8% and 32%, respectively) than in forestry sites (6% and 14%, respectively). The relative abundances of plant pathogens and saprotrophic pathogens were significantly greater in reclamation sites, while that of fungi with unknown functional properties (excluding unidentified fungi) was greater in forestry sites (*p* = 0.006; Table 2; Figure 7C). The number of taxa deemed mycoparasites and mycoparasitic yeasts was significantly greater in reclamation sites (*p* = 0.013; Table 2; Figure 7D). To a lesser extent (RDA1 < −0.27), the numbers of plant pathogens, saprotrophs, and fungi with unknown functional properties were also greater in reclamation sites (*p* = 0.013). The effects of the reclamation status without confounding factors are visualized in Appendix A.

#### 3.2.5. Sporocarp Survey

A total of 389 sporocarps of 21 fungal taxa were identified across all four jack pine transects (Gateway Hill M2 and M7 and Fort Chipewyan road TR1B2 and TR1P), consisting of 11 saprophytic, 8 EM, and 2 EM with saprotrophic capabilities, *Lyophyllum decastes* (Fr.) Singer and *Ramaria abietina* (Pers.) Quél (Appendix A). The mean numbers of saprotrophic, EM, and ecto-saprotrophic species at Gateway Hill (4, 1.5, and 1, respectively) differed from those at Fort Chipewyan road (1.5, 2.5, and 0, respectively), with more saprophytic or ecto-saprophytic sporocarps in reclamation sites and more EM sporocarps in forestry sites. The latter was mainly due to the greater sporocarp abundance and species richness in the partially burned transect (TR1P: 102 and 8, respectively) than in the severely burned transect (TR1B2: eight and two, respectively).

Across the four jack pine transects, soil DNA methods found more saprotrophic and EM (including ecto-saprotrophic) fungal taxa (pyrosequencing: 152 and 72, respectively; Illumina: 352 and 108, respectively) than the sporocarp survey (11 and 8, respectively). Not all saprotrophic and EM taxa observed in the survey were detected by pyrosequencing (missed: four and one species, respectively) and Illumina (missed: five and one species, respectively), although the two ecto-saprophytic species, *L. decastes* and *R. abientina*, were detected by both soil DNA methods. Matches to the survey at the genera level were higher for both pyrosequencing (eight saprotrophic species and seven EM species) and Illumina (nine saprotrophic species and seven EM species; Appendix A).

## 4. Discussion

### 4.1. General Differences in Community Composition and Function

The composition and functional properties of soil fungal communities were significantly different between reclamation and forestry sites. This was found in the combined 2013–2014 preliminary assessment of soil DNA using pyrosequencing, in the 2014 assessment of soil fungal communities detected by pyrosequencing and Illumina sequencing in jack pine transects, and in the sporocarp survey. Many taxa were detected only in samples from either reclamation or forestry sites, while other taxa were detected significantly more frequently in a particular site. Species-specific links between above- and below-ground communities can trigger positive or negative functional feedbacks, with serious consequences for ecosystem productivity [2]. The detected differences in taxa composition likely resulted from the origin and history of the soil material, as well as the habitat structure. Each of these factors leads to different inocula and selective pressures on fungal community development.

Material transport and human activity are important vectors of fungal inocula [58,59]. In logging operations, woody substrates for fungal colonization are removed from a site, and little or no material is imported. By contrast, surface mining followed by reclamation removes all vegetation, surface organic soil, and mineral soil from a site and imports large volumes of soil material, containing potential fungal inocula, from multiple locations. Thus, human activity facilitates opportunistic introduction and spread following reclamation, leading to legacy effects on the composition of fungal communities 25 to 30 years post-disturbance.

Jack pines do not tolerate waterlogged sites and do not grow in the acid bogs [60] where muskeg peat was sourced for reclamation efforts at Gateway Hill. Thus, many fungal taxa typically associated with jack pine soils would have been absent from this original reclamation material, while other unassociated taxa would have been introduced. Differences between the community compositions in the initial starting materials would have been compounded by selection during stockpiling, transporting, and mechanical spreading of soil material at the reclamation site. Each of these steps provided unique abiotic (e.g., oxygenation and moisture) and biotic selective pressures (e.g., competition, grazing, and symbioses), distinct from those in undisturbed sites and in sites affected by fire and tree harvesting. Further community divergence due to selection likely occurred after the site was established, as a result of peat and other soil constituents increasing water retention and/or decreasing aeration compared with soils in managed jack pine forests [60].

Most taxa were detected in more than one soil fraction, yet associations with reclamation status were not consistent across soil fractions. Varying the inoculation, dispersal, and competitive abilities of individual taxa according to the substrate type could account for some of the differences among the soil fractions. These differences among the soil fractions may be important, but our focus was to identify the systematic effects posed by reclamation status. The designation of taxa affected by the reclamation status accounted for potential confounding factors and, for pyrosequencing results, considered only taxa consistently affected across all soil fractions. We simultaneously assessed the presence/absence and relative abundance of taxa, as well as multiple indexes of diversity, to acknowledge the importance of both dominant and rare microbial species in affecting ecosystem processes, as well as to account for the limitations of semi-quantitative high-throughput sequencing. The effects of reclamation status on fungal communities in our study were therefore conservatively underestimated.

Differences in the relative abundances of functional groups were limited to mineral soils (coarse and fine fractions). The functional similarity among fungal communities in forest floors and roots, despite differences in taxa identity, may result from similar selection pressures provided by jack pine trees in both areas. Trees provide the majority of available root and litter habitats in forest systems, thereby shaping the general competitive landscape for fungi and subsequently leading to particular functional dominance (e.g., saprotrophs and ectomycorrhizae [2,8,61]). Thus, tree selection and localized functional heterogeneity masked any differences between reclamation and forestry sites in these soil fractions. By contrast, communities in mineral soils were under less convergent selective pressure from trees and therefore more drastically demonstrated differences in functional properties associated with reclamation status.

### 4.2. Greater Fungal Richness and Diversity in Reclamation Soils

The greater fungal richness and diversity at the reclamation site may have been artefacts of historical legacies or alternatively may have resulted from greater niche differentiation, allowing for localized competitive success by different taxa according to individual traits (e.g., foraging strategies and stress tolerance). Functional redundancy among fungal taxa tends to reduce diversity due to niche saturation at low species richness and to inter-specific competitive exclusion [59]. Compared with soils of logged, fire-disturbed, and intact forests, the seemingly homogenized soil habitat at the reclamation site may thus foster a greater heterogeneous mosaic of distinct niches than is apparent. The niche richness at the reclamation site could result from exposures to elevated salt and hydrocarbon levels in overburden [29], which were not measured in our study. Alternatively, the mixing of muskeg peat and sandy loam could have led to a lasting mosaic of habitats in topsoils, each substrate providing different environmental conditions, selecting for distinct community complexes from initial inocula.

Fungal richness was not associated with tree species at our sites, similar to results found in other high-throughput sequencing studies of soil systems [7,62]. This contrasts with more detailed soil DNA studies of one root and three soil fractions in different species stands at Gateway Hill in 2014 and 2016, which found fungal richness per trophic groups varied with tree species, more ectomycorrhizal taxa being found in spruce stands, more saprotrophic taxa in pine stands, and more pathogenic taxa in the larch stand [63]. Nonetheless, the identity of plant litter can affect the saprotroph composition and, in some cases, richness, in mixed-temperate forests [64]. Scant understory cover at reclamation and forestry sites would have reduced the possibility of plant litter grossly affecting the saprotroph richness. Our study did not explicitly account for differences in understory cover, so these cannot be decisively ruled out as potential contributors to the fungal richness.

Not surprisingly, more taxa were detected through soil DNA methods than from the sporocarp survey, which relies on the observation of ephemeral fruiting structures. However, both showed higher total richness at the reclamation stands. The absence of taxa detected by the sporocarp survey from lists of taxa detected by metagenomics approaches might be due to limited soil DNA sampling, differences in informatics processing of sequence data, or limitations in fungal DNA databases.

### 4.3. Ectomycorrhizae Composed a Greater Proportion of Fungal Taxa Associated with Forestry Soils

In the preliminary assessment and jack pine transects, about half of the saprotrophic and EM taxa detected by soil DNA were significantly associated with either the reclamation area or the forestry area, inferring potentially distinct ecosystem functioning between locales. Jack pines at reclamation and forestry sites likely have a comparable degree of EM root colonization with a similar number of taxa, albeit different species. Nonetheless, EM fungal taxa constituted a greater proportion of taxa at the forestry than at the reclamation jack pine transects according to the sporocarp survey (62% versus 40%) and of area-associated taxa detected by pyrosequencing (25% versus 16%) and Illumina sequencing (33% versus 12%). A similar trend was observed at a different reclaimed site in Alberta where the undisturbed forest contained primarily EM fungi, while the adjacent reclaimed area was dominated by arbuscular mycorrhizae [65]. Symbiotic relationships with EM fungi are not all equal and can range from mutualistic to quasi-parasitic [66]. Co-adaptations between trees and soil microorganisms, including EM symbionts, can be genotype [67,68]. In addition to the species specificity of EM associations, the shift from beneficial to detrimental symbioses can be context-specific [59]. Despite community differences, *Amphinema byssoides* (Pers.) J. Erikss., Sebacina spp. and *Wilcoxina* spp. (including *W. rehmii*) were abundantly detected in both areas and are known EM inoculants of pine roots in reclamation areas (Bois et al., 2005 [26,69]). Yet *Sebacina vermifera* Oberw. was more often associated with reclaimed soils, while some *Amphinema* species were preferentially associated with forestry soils.

Tree species are also usually well adapted to distinct soil nutrient cycles [70,71], which are regulated, in part, by the available saprotrophs [1,72]. We did not measure the microbial activity and nutrient flows at our sites and hence cannot determine the extent of functional similarity between the distinct saprotrophic and EM communities at reclamation and forestry sites and how this extent changes after disturbance. MacKenzie and Quideau [73] examined three recent oil sands reclamation sites planted with *Populus tremuloides* and *Picea glauca* into a surface-placed peat/mineral soil mix three years earlier. They found microbial communities varied most by seasonal soil moisture and then by site, while NH_4_-N and NO_3_-N varied most by site, reflecting differences among the source materials of soil mixes and suggesting that there was a linkage between the microbial community structure and nutrient availability three years post-reclamation. Furthermore, some saprotrophs, such as *Phialocephala* species, have endophytic life stages that can affect the health of their tree hosts [74]. Dark septate fungi, which include *Phialocephala* species and a complex of associated species [75], have previously been reported as a potential inoculant of pine roots in muskeg-peat-containing substrates [26]. Our findings therefore warrant further investigation into the relationships between tree stocks and available soil fungal inocula for the long-term productivity and resilience of forests regrowing on mined sites.

### 4.4. More Taxa with Unknown Functional Properties Were Detected in Reclamation Soils

Globally, less than approximately 10% of existing fungi are estimated to be taxonomically identified [76], let alone functionally characterized. Similar to the results of a global study of soil fungi [7], more than one-fourth of the taxa detected in our study were not associated with functional groups, and the majority of these taxa did not match known GenBank taxa. The greater number of these taxa detected in reclamation soils may reflect the greater detected richness at the Gateway Hill reclamation site. Nonetheless, the high abundance of unknown taxa underlines the lack of knowledge regarding the diversity and ecosystem dynamics in such systems. The greater number of functionally unknown taxa in soils from reclamation sites could further underline fundamental differences in nutrient cycling, fungal symbioses, or pathogenic pressures, each of which could threaten long-term forest productivity and resilience.

### 4.5. Pathogens Were More Abundant and Richer in Taxa in Reclamation Soils

The ubiquitous presence of *Beauveria bassiana* (arthropod parasite) and the greater number and abundance of plant pathogens in reclaimed area soils could have resulted from the greater prevalence of these taxa in the starting material and their selective survival during storage due to resistant resting stages, such as sclerotia. Sclerotia formation is a common trait in many necrotrophic plant pathogens, such as *Leptosphaeria sclerotioides* (Preuss ex Sacc.) Gruyter, Aveskamp & Verkley and *Verticillium* spp. [24]. Alternatively, destabilization of ecosystem dynamics could have facilitated the inoculation by and spread of opportunistic pathogens [56] during soil storage and transport or following the site re-establishment. The exposure of plants to pathogens with little coevolutionary history could pose a challenge to long-term species or ecosystem resilience [77].

The direct threat posed by detected pathogens to forests at the reclamation site is unclear. Many detected pathogens were known generalist pests of typically agricultural crops (e.g., *Leptosphaeria sclerotioides*, *Verticillium* species, and *Olpidium brassicae* (Woronin) P.A. Dang.), but they may also target trees and affect ecosystem dynamics. *Olpidium brassicae* has previously been shown to infect pine seedlings planted in oil sands reclamation areas [69]. Although pine seedlings were observed to be more abundant on the TR1 vs. GH jack pine transects, it did not mirror trends in plant pathogen abundance in our study. However, these relationships could not be explicitly tested. Nonetheless, links between understory plants and plant pathogen diversity are typically limited to host-specific pests [77,78,79]. A *Colpoma* species that most closely matched the New Zealand collection PDD 91607 was predominantly found in the jack pine root-containing samples at the forestry sites, yet was not restricted to this species or location. There are 14 species of *Colpoma* with a widespread distribution [80]. *Colpoma crispum* (Pers.:Fr) Sacc. has been collected from *Tsuga heterophylla* (Raf.) Sarg., *Pinus monticola* Douglas ex D. Don, and *Pseudotsuga menziesii* (Mirb.) Franco in British Columbia [81]. *Colpoma quercinum* (Pers.) Wallr., an endophyte of oak, has been isolated from twigs of healthy and declining oaks [82] and in dead roots [83] of *Quercus robur* Linnaeus. We did not observe *Colpoma* spp. during our sporocarp survey, and it is difficult to know whether it was present in root pieces or small wood fragments present in the soil, but its presence at multiple sites indicates that the soil DNA techniques we employed were able to detect inconspicuous fungi even when they were not fruiting.

### 4.6. Accounting for Potentially Confounding Environmental Factors

The effects of reclamation status on fungal communities (outlined above) were determined using multiple statistical approaches, two high-throughput sequencing platforms, a sporocarp survey, and two sampling designs, each accounting for different confounding factors. The geospatial separation of reclamation and forestry sites, as well as differences in mineral soil P and forest floor depths, was omitted from our analysis of reclamation effects. The effects of these potentially confounding factors were likely correlated with those of the reclamation status and not actually causal.

The forest floor depth was significantly associated with fungal communities in mineral soils and root fractions and less so with communities in forest floors. The causes of these associations are unlikely to be biological but may be indirectly linked to fire disturbance. The fire history at the Fort Chipewyan road forestry sites can explain the different forest floor depths and mineral soil P concentrations (and to a lesser extent the total N in mineral soils) in jack pine stands where these variables were measured. The 2011 Richardson fire burned the accumulated forest floor layer, releasing the P bound in organic matter and allowing it to leach to mineral soils, while some of the stored N was volatized [84]. The fire intensity affected the composition of fungal communities by favoring heat-tolerant, early-successional pioneer species. This was supported by soil DNA and sporocarp surveys at the Fort Chipewyan transects, where EM fungal taxa in the severely burned stand were depauperate when compared with the partially burned stand. However, the saprophytic and overall fungal richnesses were similar. Nonetheless, the effects of reclamation status were not confounded by fire intensity and were consistent when unburned forestry stands were included. The management of resource extraction operations such as logging often attempts to reduce environmental impacts by mimicking natural disturbance cycles such as fire [12]. Our results show greater similarity between the impacts of recent logging and fire disturbance on soil fungal communities compared with the impacts of oil sands mining 25 to 30 years after reclamation and revegetation.

Greater soil P concentrations are known to occur in ecosystems under more frequent fire regimes and have been associated with less fungal richness at global scales [7]. Nonetheless, all samples showed stoichiometric limitations of P on general fungal growth [85]. The biological causes behind the significant associations between fungal communities in forest floors and P concentrations in mineral soils at our sites are unclear, especially since more fungal species were found in the reclamation sites, which had greater P limitation than forestry sites. Thus, significant associations with P concentrations may simply reflect the correlated effects of reclamation status, similar to the effects of forest floor depth.

Increasing the spatial distance, even within the same ecosystem, generally increases the dissimilarity among soil microbial communities [86,87,88]. Accounting for the spatial layout is therefore important when undertaking analyses of microbial communities in field settings. Unfortunately, the available sampling locations prevented us from thoroughly separating the effects of spatial autocorrelation from the effects of reclamation status (i.e., PCNM 1). The geographic ranges of fungal taxa tend to increase with latitude [7]. The relatively small distance (considering the vast boreal ecosystem) and lack of major dispersal barriers between the Gateway Hill reclamation and the Fort Chipewyan road forestry sites should therefore reduce the predominance of spatial separation effects. Decisively separating spatial effects from land reclamation would require laying out an experimental design at the time of the initial site disturbance followed by monitoring and maintenance of the research plots over a course of over 40 years (for data comparable to those analyzed in this study). Although challenging, randomized block designs testing the effects of resource extraction operations on whole-ecosystem processes (including soil communities) is needed for better assessments of the long-term impacts.

## 5. Conclusions

The composition and richness of fungal communities in soils at reclamation and forestry sites are distinct and are likely the result of differences in inocula and selective pressures. Saprotrophic, EM, and plant pathogenic fungi composed the bulk of the community differences between the Gateway Hill and Fort Chipewyan road areas and have the potential to critically affect the productivity and resilience of reclaimed forests. Our hypothesis that fungal richness and diversity would be greater in the forestry sites was not supported, as greater richness and number of unknown taxa were found in reclaimed forest soils, suggesting high habitat complexity with understudied biota. However, our hypothesis that the EM fungi would account for the largest differences between forestry and reclamation sites was supported as all methods found that EM species abundance was greater in forestry sites. To reduce the potential for mismatch between above- and below-ground communities, we suggest ecosystem recovery efforts consider the similarity of the microbiome between the source material and the target forest systems. Our findings warrant further long-term research into fungal biogeography and associated ecosystem processes in experimentally controlled reclamation areas, as well as nearby and regional managed and unmanaged forests.

## Figures and Tables

**Figure 1 jof-09-01110-f001:**
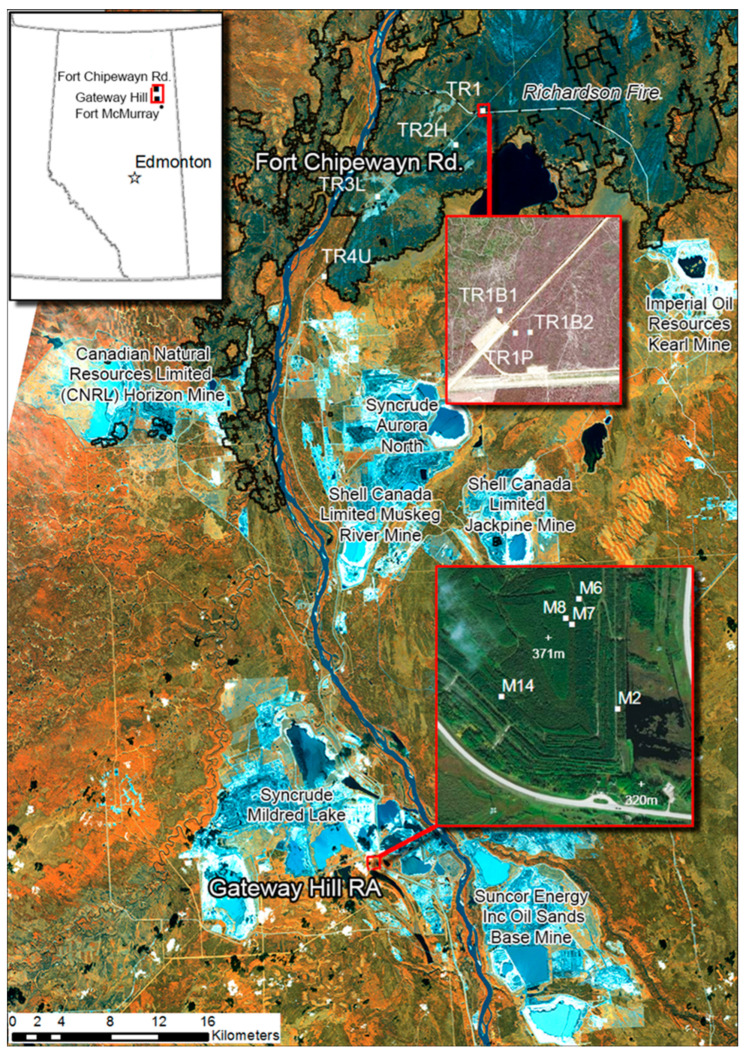
Locations of sampled reclaimed oil sands forest sites at the Gateway Hill Reclamation Area (RA) and forestry sites along the Fort Chipewyan road area within the bitumen mines region north of Fort McMurray, Alberta. See Table 1 for stand descriptions. Sources: Main map—Rapid Eye (acquired 6 September 2012) displayed as NIR, Red Edge and Red in RGB. Blue—unvegetated area such as mines; black—water bodies and cloud shadows; orange—deciduous dominated vegetation; olive green—conifer-dominated vegetation. Richardson 2011 fire boundary derived from interpreted 2012 Landsat composite [27]. Fort Chipewyan Rd. TR1 inset map—2014 Spot6 imagery with pre-fire AVI vegetation cover polygons, courtesy B. Glover, Alberta Dept. Agriculture and Forestry. Gateway Hill RA inset map—2007 imagery from ESRI, Digital Globe, Earthstar Geographics, CNES/Airbus DS, UEDA, USGS, AeroGrid, IGN, and the GI User Community. Cartography by A. Dyk, Canadian Forest Service, Natural Resources Canada, Deforestation monitoring group.

**Figure 2 jof-09-01110-f002:**
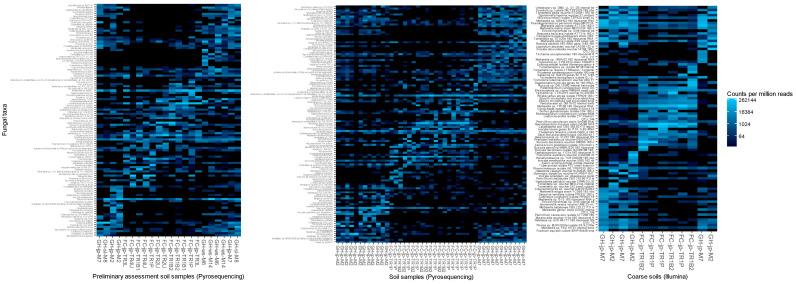
Dissimilarity analyses of the relative abundances of soil fungal taxa detected by pyrosequencing or Illumina sequencing in jack pine (jp), white spruce (ws), and Siberian larch (sl) stands from the Gateway Hill (GH) and Fort Chipewyan road (FC) areas. Analyses of fungal taxa with variance in relative abundance greater than 1 × 10^−5^ (109 taxa for preliminary assessment all stands, 120 taxa for pyrosequenced taxa in jack pine transects and 87 taxa for Illumina sequencing jack pine stand transects). Heat maps display Bray–Curtis community dissimilarity using non-metric multidimensional scaling (NMDS) ordering of samples on the x-axis and displaying a log_4_ color scale. The sample order was automatically generated based on the radial coordinate angles of ordination axes 1 and 2.

**Figure 3 jof-09-01110-f003:**
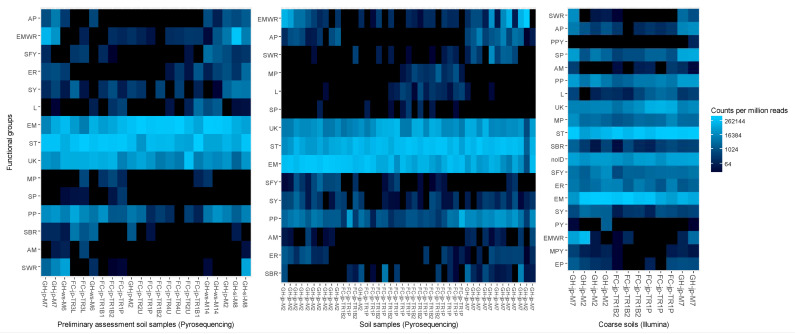
Dissimilarity analyses of the relative abundances of soil fungi functional groups detected by pyrosequencing or Illumina sequencing in jack pine (jp), white spruce (ws), and Siberian larch (sl) stands from the Gateway Hill (GH) and Fort Chipewyan road (FC) areas for all stands in the preliminary assessment and for the jack pine only transects. Heat maps display Bray–Curtis community dissimilarity using non-metric multidimensional scaling (NMDS) ordering of samples on the x-axis and displaying a log_4_ color scale. The sample order was automatically generated based on radial coordinate angles of ordination axes 1 and 2. UK, unknown function; EM, ectomycorrhizal; AM, arbuscular mycorrhizal; PP, plant pathogen; ST, saprotroph; SY, saprotroph yeast; SFY; saprotroph facultative yeast; SBR, saprotroph brown rot; L, lichenized; SWR, saprotroph white rot; AP, animal parasite; ER; ericoid; MP, mycoparasite; EP, endophyte; SP, saprotroph pathogen; noID, fungal taxa not identified; EMWR, ectomycorrhizal white rot.

**Figure 4 jof-09-01110-f004:**
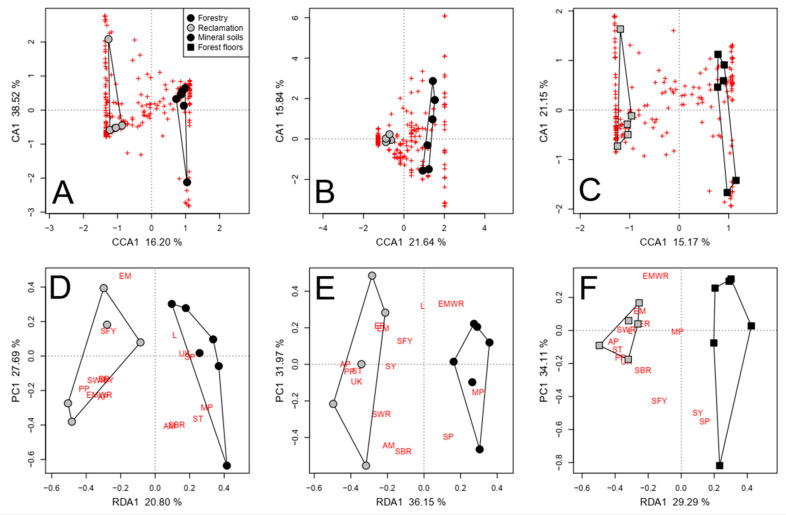
The direct effects of reclamation status (symbol color = stand type, symbol shape = soil fraction), as depicted by partially constrained ordinations of the relative abundances (**A**,**C**) and presence/absence (**B**) of pyrosequenced fungal taxa, the relative abundance of functional groups (**E**), and the number of fungal taxa belonging to functional groups (**D**,**F**) in preliminary assessments of mineral soils (**A**,**B**,**D**,**E**) and forest floors (**C**,**F**), while removing respective variance significantly associated to confounding factors (i.e., spatial components other than PCNM1). Both species and site scores represent unscaled raw eigenvalues. Polygons represent k-means clustering based on axes 1 and 2. Functional group abbreviations in (**D**–**F**) are defined in the Figure 3 caption. Each red cross represents a unique pyrosequenced fungal taxon.

**Figure 5 jof-09-01110-f005:**
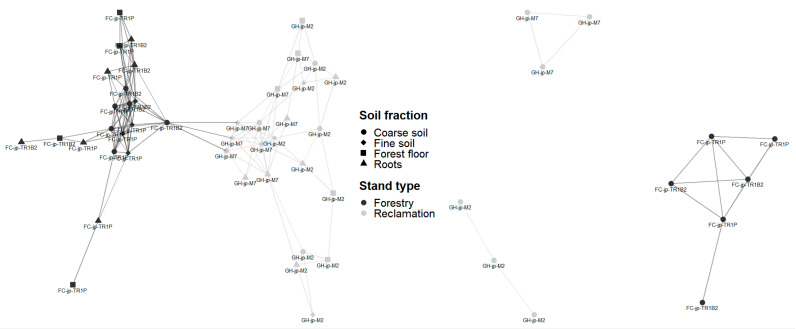
Network analysis of the relative abundance of fungal taxa detected by pyrosequencing (**left**) or Illumina sequencing (**right**) in samples from the Gateway Hill (GH) and Fort Chipewyan road (FC) jack pine transects. The distance threshold of 0.71 using the Bray–Curtis index was the smallest index that displayed links in all 12 Illumina-sequenced samples and 42 of 43 pyrosequenced samples (i.e., forest floor replicate 3 at M7 is not displayed).

**Figure 6 jof-09-01110-f006:**
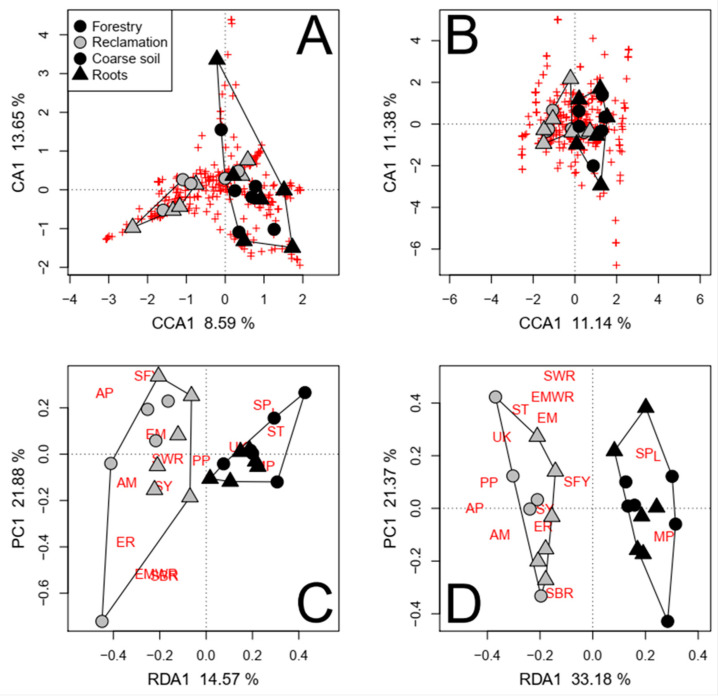
The direct effects of reclamation status (symbol color = stand type, symbol shape = soil fraction), as depicted by partially constrained ordinations of the relative abundances and presence/absence of pyrosequenced fungal taxa ((**A**,**B**), respectively), the relative abundance of functional groups (**C**), and the number of fungal taxa belonging to functional groups (**D**) in roots and coarse soils in jack pine transects, while removing the effects of soil fraction and respective variance significantly associated to confounding factors (regeneration year, NO_3_, NH_4_, total N, and/or PCNM 2). Both species and sites scores represent unscaled raw eigenvalues. Polygons represent k-means clustering based on axes 1 and 2. Each red cross represents a unique pyrosequenced fungal taxon. The functional group abbreviations in (**C**,**D**) are defined in the Figure 3 caption.

**Figure 7 jof-09-01110-f007:**
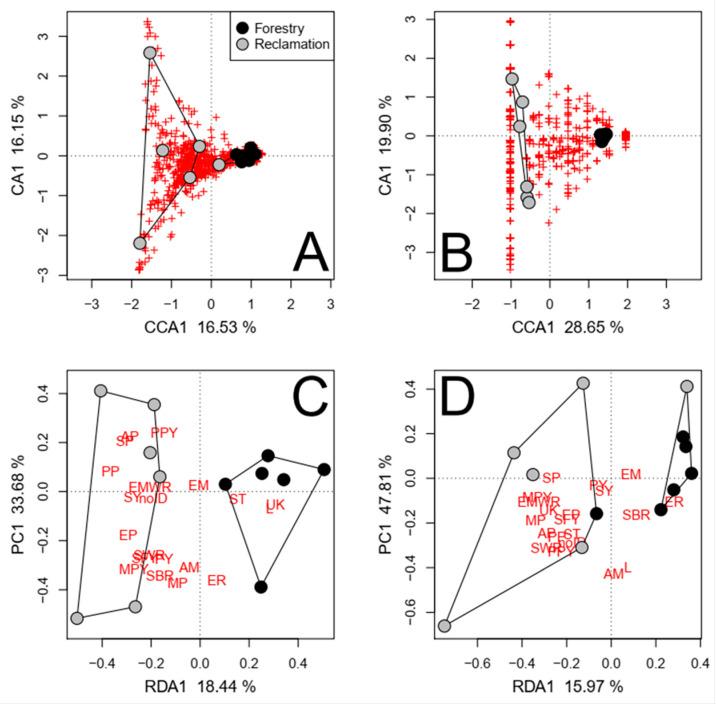
The direct effects of the reclamation status as depicted by partially constrained ordinations of the relative abundances and presence/absence of Illumina-sequenced fungal taxa ((**A**,**B**), respectively), the relative abundance of fungal trophic functions (**C**), and the number of fungal taxa belonging to fungal trophic functions (**D**) in coarse soils in jack pine transects, while removing the variance associated to the respective significant confounding factors (regeneration year, NO_3_, NH_4_, total N, PCNM 2, and/or PCNM 3). Both species and sites scores represent unscaled raw eigenvalues. Polygons represent k-means clustering based on axes 1 and 2. Each red cross represents a unique pyrosequenced fungal taxon.

**Table 1 jof-09-01110-t001:** Descriptions and geospatial coordinates of sites sampled at the Gateway Hill (GH) and Fort Chipewyan (FC) road areas.

Area	Site	Tree Species	Latitude (N)	Longitude (W)	Soil Placed	Regeneration Year	Disturbance Type	Samples 2013–2014	Samples 2014	Forest Floor Depth (cm)
FC	TR1B1	Jack pine	57.54356	111.36559	NA	2012	Severe burn	1		0.2
FC	TR2H	Jack pine	57.51772	111.40374	NA	1996	Harvest	1		1.5
FC	TR3L	Jack pine	57.48267	111.51478	NA	1954	Light partial burn	1		1.5
FC	TR4U	Jack pine	57.42612	111.59225	NA	1950	Undisturbed	1		20.0
FC	TR1B2	Jack pine	57.54317	111.36305	NA	2012	Severe burn	1	3	0.2
FC	TR1P	Jack pine	57.54206	111.36395	NA	1954	Partial burn	1	3	0.8
GH	M6	White spruce	56.99657	111.56528	1990	1990	Reclamation	1		2.4
GH	M14	White spruce	56.99309	111.57028	1984	1985	Reclamation	1		3.5
GH	M8	Siberian larch	56.99588	111.56616	1990	1990	Reclamation	1		5.3
GH	M2	Jack pine	56.99269	111.56274	1982	1983	Reclamation	1	3	1.8
GH	M7	Jack pine	56.99566	111.56574	1990	1990	Reclamation	1	3	3.4

NA, not applicable.

**Table 2 jof-09-01110-t002:** Variance partitioning (%) between reclamation status and other significant confounding factors for (A) fungal taxa (relative abundance and presence/absence) or (B) functional groups (relative abundance and number of taxa belonging to each group) in jack pine transects. Community data from coarse soils, fine soils, roots, and forest floors were treated separately and/or jointly.

Response	Soil Fraction	Reclamation Status	Shared	Other Factors	Residual
(A) Fungal taxa					
Relative abundance					
Coarse soils ^†^		16.53 ***	1.78	10.91	70.78
Coarse soils ^‡^		18.14 **	0	13.29	68.57
Fine soils ^‡^		18.58 ***	0.07	14.10	67.25
Forest floors ^‡^		15.86 **	1.02	14.62	68.50
Roots ^‡^		13.22 **	0	0	86.78
Both roots and coarse soils ^‡,§^		8.59 ***	2.23	19.61	69.57
All soil fractions ^‡,§^		5.56 ***	2.20	14.40	77.84
Presence/absence					
Coarse soils ^†^		28.65 **	0	0	71.35
Coarse soils ^‡^		21.94 **	0	0	78.06
Fine soils ^‡^		21.21 **	5.16	14.32	59.31
Forest floors ^‡^		16.61 *	2.96	27.83	52.60
Roots ^‡^		15.20 ***	1.47	10.35	72.98
Both roots and coarse soils ^‡,§^		11.14 ***	4.25	19.72	64.89
All soil fractions ^‡,§^		7.33 ***	6.21	21.51	64.95
(B) Functional groups					
Relative abundance					
Coarse soils ^†^		18.44 **	0	34.65	46.91
Coarse soils ^‡^		30.05 ***	0	18.10	51.85
Fine soils ^‡^		17.45 *	6.97	29.84	45.74
Forest floors ^‡^		15.36	0	22.82	61.82
Roots ^‡^		11.53	0.54	13.73	74.20
Both roots and coarse soils ^‡,§^		14.57 ***	0	20.18	65.25
All soil fractions ^‡,§^		9.33 ***	0.87	10.59	79.21
Number of taxa					
Coarse soils ^†^		15.97 *	21.16	16.46	46.41
Coarse soils ^‡^		46.46 **	0	0	53.54
Fine soils ^‡^		34.19 **	1.25	16.28	48.28
Forest floors ^‡^		11.63	16.88	16.01	55.48
Roots ^‡^		30.91 ***	0	0	69.09
Both roots and coarse soils ^‡,§^		33.18 ***	0	13.70	53.12
All soil fractions ^‡,§^		25.82 ***	0.43	10.19	63.56

* Significant reclamation status effects at the 0.05 probability level. ** Significant reclamation status effects at the 0.01 probability level. *** Significant reclamation status effects at the 0.001 probability level. ^†^ Communities sequenced using Illumina sequencing. ^‡^ Communities sequenced using pyrosequencing. ^§^ Models were each conditioned by the soil fraction.

## Data Availability

Files for pyrosequencing, Illumina, and environmental data for each sample and R scripts used in the statistical analyses are available on GitHub. https://github.com/patg13/reclamation_site_data (accessed on 1 November 2023).

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
