# Peer review of "Differences in Soil Fungal Communities between Forested Reclamation and Forestry Sites in the Alberta Oil Sands Region"

_jof, 2023, doi:10.3390/jof9111110_

Round 1

Reviewer 1 Report

Comments and Suggestions for Authors

The authors of the paper "Differences in soil fungal communities between forested reclamation and forestry sites in the Alberta oil sands region" available to me have written a very interesting article.

Nevertheless, I have a few comments and tips that will hopefully improve the quality of the paper.

In principle, this paper is exorbitantly long. I can very well imagine the authors' internal balancing act between a detailed presentation of all the necessary results and a focus on the most essential content……….. In this context, I would also include the current tables 3 and 4, each of which extend over several pages to the supplementary files.

Here are a few smaller, technical things that I noticed:

·         In the site description I am missing a climatic classification of the region under investigation. This is particularly important for international readers.

·         All scientific names in the body text must be written in italics.

·         The order of the test areas shown in L. 137 does not make sense to me, especially since it was not shown that way in Table 1.

·         L. 144: If this stock was established in 1950, it is now 73 years old and not 60 years old.

·         L. 186 I didn't find a table B.1, did you mean S.1?

·         The abbreviations for ammonium and nitrate must be written correctly (with superscripts and subscripts).

·         The axis labels in Fig. 2 cannot be read without aids; the same applies to Fig. 3.

·         The significances must be represented with the special character ≤.

·         The font in Fig. 5 is much too small!

Author Response

Reviewer 1

The authors of the paper "Differences in soil fungal communities between forested reclamation and forestry sites in the Alberta oil sands region" available to me have written a very interesting article.

Nevertheless, I have a few comments and tips that will hopefully improve the quality of the paper.

In principle, this paper is exorbitantly long. I can very well imagine the authors' internal balancing act between a detailed presentation of all the necessary results and a focus on the most essential content……….. In this context, I would also include the current tables 3 and 4, each of which extend over several pages to the supplementary files.
Tables 3 and 4 moved to Supplementary Material, Tables S6 and S9 respectively.

Here are a few smaller, technical things that I noticed:

  • In the site description I am missing a climatic classification of the region under investigation. This is particularly important for international readers.
    Site and climatic descriptions added to the general description of the area in the Introduction at line 105.
  • All scientific names in the body text must be written in italics.
    Corrections made to show all species names in italics. All were italicized in the original manuscript but when reformatted to the journal style, italics were lost.
  • The order of the test areas shown in L. 137 does not make sense to me, especially since it was not shown that way in Table 1.
    Line 137 and rest of section 2.1 revised to indicate forest cover type in the two areas as shown in Figure 1.
    The order of sites listed in Table 1 revised to match the sites as given in text of section 2.2.
  • L. 144: If this stock was established in 1950, it is now 73 years old and not 60 years old.
    Corrected to ~70-year-old stand. Stand age is from a forest cover map and may not reflect tree ages.
  • L. 186 I didn't find a table B.1, did you mean S.1?
    Corrected to refer to Table S1
  • The abbreviations for ammonium and nitrate must be written correctly (with superscripts and subscripts).
    Corrections made to abbreviations for ammonium and nitrate throughout manuscript
  • The axis labels in Fig. 2 cannot be read without aids; the same applies to Fig. 3.
    We understand the comment, but unfortunately cannot change the figure as it is a direct representation of the dissimilarity analysis performed. Despite its small size, the font is set to the largest size that does not cause overlapping labels. Individual taxon labels are legible when zooming into the document in word format (the publisher should ensure this is the case in the published pdf version of the manuscript). We have uploaded to the journal high resolution pdf versions of Figure 2 and 3 to ensure these figures that allow for zooming in.
  • The significances must be represented with the special character ≤.
    Changes were made and special symbol used in lines 371, 397, 449. Significances are reported to 3 decimal places and so those <0.001 cannot necessarily be reported as ≤ 0.001.
  • The font in Fig. 5 is much too small!
    We have revised the figure to use the largest font size that does not cause too much overlapping labels. The symbol legend in this new figure now also has a larger font size. The individual site names are legible when the zoom is feature is used.

Reviewer 2 Report

Comments and Suggestions for Authors

Thank you for the manuscript "Differences in soil fungal communities between forested recla- 2 mation and forestry sites in the Alberta oil sands region." It provides an interesting analysis of fungal communities in different ecosystems.

I would have the following suggestions: 

1) Figure 2: Please make the image more readable and clear. The taxa on the y-axis are not clear. Maybe reduce the number of taxa? Please present a clustering of the taxa on y so the reader can spot patterns easily. In this case, you can also include two colors for plotting, not just blue. Make the color scheme clearer

2) Please share the raw data and scripts you have used for the analysis

Minor

1) Check for species names that need to be in italics (ex. line 360) 

Author Response

Reviewer 2

Thank you for the manuscript "Differences in soil fungal communities between forested reclamation and forestry sites in the Alberta oil sands region." It provides an interesting analysis of fungal communities in different ecosystems.

I would have the following suggestions: 

1) Figure 2: Please make the image more readable and clear. The taxa on the y-axis are not clear. Maybe reduce the number of taxa? Please present a clustering of the taxa on y so the reader can spot patterns easily. In this case, you can also include two colors for plotting, not just blue. Make the color scheme clearer
We understand the comment, but unfortunately cannot change the figure as it is a direct representation of the dissimilarity analysis performed. Hence, we cannot reduce the number of taxa when plotting as this would potentially alter the clustering of soil samples on the x-axis and mislead readers. The clustering of taxa on the y-axis is already automatically optimized by the plotting function. We have tested various colour schemes and feel as thought the blue fading to black gives the best balance of both contrast for the viewer and meaning (i.e., black represents the absence of a taxon). Despite its small size, the font is set to the largest size that does not cause overlapping labels. Individual taxon labels are legible when zooming into the document in word format (the publisher should ensure this is the case in the published pdf version of the manuscript). We have uploaded to the journal high resolution pdf versions of Figure 2 and 3 to ensure these figures that allow for zooming in.

2) Please share the raw data and scripts you have used for the analysis
The scripts for the statistical analyses are all in R, quite streamlined, and not coded in a user-friendly fashion as it includes code for exploratory analyses which likely has no value to other users. We have contacted the JoF journal editors to identify where the data sets and scripts for the data analyses can be uploaded and will add a link in the Supplemental material as to where these are posted.  We will also provide a link to the DNA fasta files from this study when they have been archived for online access.

Minor

1) Check for species names that need to be in italics (ex. line 360) 
Corrections made to show all species names in italics. All were italicized in the original manuscript but when reformatted to journal style, italics were lost.